# ORCHID: A Chinese Debate Corpus for Target-Independent Stance Detection and Argumentative Dialogue Summarization

**Xiutian Zhao**[†]   **Ke Wang**[†]   **Wei Peng**[*]
Huawei IT Innovation and Research Center
{zhaoxiutian, wangke215, peng.wei1}@huawei.com

## Abstract

Dialogue agents have been receiving increasing attention for years, and this trend has been further boosted by the recent progress of large language models (LLMs). Stance detection and dialogue summarization are two core tasks of dialogue agents in application scenarios that involve argumentative dialogues. However, research on these tasks is limited by the insufficiency of public datasets, especially for non-English languages. To address this language resource gap in Chinese, we present OR-CHID (**Or**al **Chi**nese **D**ebate), the first Chinese dataset for benchmarking target-independent stance detection and debate summarization. Our dataset consists of 1,218 real-world debates that were conducted in Chinese on 476 unique topics, containing 2,436 stance-specific summaries and 14,133 fully annotated utterances. Besides providing a versatile testbed for future research, we also conduct an empirical study on the dataset and propose an integrated task. The results show the challenging nature of the dataset and suggest a potential of incorporating stance detection in summarization for argumentative dialogue.[1]

## 1  Introduction

Recent development of large language models (LLMs) have pushed the general interest on dialogue agents to a new level, and increasingly powerful LLMs such as GPT series demonstrated promising capabilities across multiple application scenarios. Among various tasks that have been assigned to dialogue agents, engaging argumentative dialogues (Macagno, 2000; Walton, 2008) has long been a challenging one. Regardless of specific aims, whether winning a debate (Zhang et al., 2016), convincing people (Prakken et al., 2020), or

opening up minds (De Kock and Vlachos, 2021; Farag et al., 2022), dialogue agents rely on two of foundation abilities: stance detection and summarization. Stance detection aims to reveal attitudes of arguments, and the goal of summarization is to collect and condense information in order to build arguments. They collaboratively support comprehending and developing arguments, consequently both abilities are crucial for engaging argumentative dialogues (Chen et al., 2017; Lawrence and Reed, 2019; Wang and Wan, 2021; Sanders et al.,

Figure 1: An excerpt of one debate in ORCHID. One debate entry of our dataset consists of: (1) debate topic, (2) position statements of both sides, (3) utterances labelled with speaker and stance, and (4) stance-specific summaries. Original text is in Chinese (see Appendix G for a more complete example).

---

[†]Equal contribution.
[*]Corresponding author.
[1]ORCHID is publicly available at https://github.com/xiutian/OrChiD

2022).

In general natural language processing (NLP), stance detection is to classify the stance ('favor', 'against' and 'none') of a piece of comment with respect to a target entity or claim (Hasan and Ng, 2013; Nguyen et al., 2016; Küçük and Can, 2020; Hardalov et al., 2022). The other task, text summarization aims to compress information from a large piece of text and produce a concise and comprehensible digest (Gillick et al., 2009; Shang et al., 2018). Dialogue summarization, fittingly, takes dialogues as source text.

However, some unique features of **argumentative dialogues** propose atypical challenges for the two tasks. Regarding summarization tasks, argumentative dialogues, such as debates, meetings and online forum discussions, often contain contradictory utterances with conflicting stances (Zou et al., 2021), making them more convoluted to summarize. Also, in comparison with written text, spoken dialogues naturally carry more noises such as mispronounces, rephrasing and repeated words that obstruct summarizaiton. Meanwhile, unlike typical target-specific stance detection whose targets are explicit entities (e.g., 'Metaverse'), stance detection on argumentative dialogues is **target-independent**, meaning that the targets of those studies are claims in the form of complete sentences (e.g., 'The commercial value of metaverse is overestimated').

Despite the progress made on the tasks, research community's effort has been decelerated by an insufficiency of proper language resources. Existing dialogue summarization datasets are dominantly in English (Gliwa et al., 2019; Durmus and Cardie, 2019; Roush and Balaji, 2020; Fabbri et al., 2021; Chen et al., 2021). Regarding non-English summarization datasets, prior resources in Chinese are either focus on one specific domain (Song et al., 2020; Zou et al., 2021; Lin et al., 2021; Huang et al., 2020), or without stance-specific summaries (Feng et al., 2022). There still is a lack of multi-domain and annotated Chinese dialogue summarization datasets. Moreover, most existing stance detection datasets are target-specific regardless of language (Alturayeif et al., 2023). Overall, in terms of Chinese language resources, the amount of datasets suitable for argumentative dialogue summarization is highly limited, and there currently is no benchmark for target-independent stance detection.

To remedy this shortage and facilitate related research, we present ORCHID (**Or**al **Chi**nese **De**bate), to the best of our knowledge, the first Chinese debate summarization dataset annotated with multi-granularity and stance-specific summaries, also the first Chinese benchmark for target-independent stance detection. Our dataset consists of 14,133 fully annotated utterances and 2,436 stance-specific summaries from 1,218 real-world debates that were conducted in Mandarin. We employed an automatic speech recognition (ASR) to transcribe raw data, followed by manual post-correction and annotation. We provide debate summaries on two-levels of granularity, short concise statements and long comprehensive summaries of both stances. Stances and debaters are labelled at utterance level. Furthermore, we conduct a preliminary empirical study on the constructed data. Sharing this novel dataset, we hope to resupply the research community tackling tasks including dialogue summarization, stance detection and other argument mining tasks.

In summary, our contributions are three-fold: **(1)** we introduce ORCHID, the first Chinese dataset for debate summarization and target-independent stance detection; **(2)** we propose a new integrated task, stance-specific summarization, which is suggested by the experiment results to improve summarization on argumentative dialogues; and **(3)** we conduct preliminary experiments, benchmarking classical and newly-suggested tasks against our dataset, reporting corresponding results, and setting initial baselines for future work.

## 2   Related Work: Existing Datasets

We reviewed existing dialogue summarization and stance detection datasets as presented in Table 1 and 2. For dialogue summarization datasets, we observe a major amount imbalance between argumentative ones and non-argumentative ones. Also, argumentative dialogue summarization corpora are primarily meeting transcripts (Kumar and Kabiri, 2022), and non-English ones are rare. Stance detection studies suffer from a lack of stance detection datasets in Chinese, in particularly target-independent ones.

### 2.1   Stance Detection Datasets

Most previous stance detection studies can be grouped into two categories by target-comment dependency: (1) target-specific and (2) target-independent (Küçük and Can, 2020). While much

| Dataset | Lg. | # Dialogue | Content Domain | Content Type | Argumen-tative | Dialogue Topic | Summary Types |
|---|---|---|---|---|---|---|---|
| SAMSum | En | 16,369 | multiple | fictitious chat logs | ✗ | ✗ | abs. |
| MSAMSum | 4[a] | 5.930 | multiple | fictitious chat logs | ✗ | ✗ | abs. |
| CSDS | Zh | 10,701 | single | customer service logs | ✗ | ✓ | abs. & role-oriented |
| SportsSum | Zh | 5,428 | single | sports commentaries | ✗ | ✓ | abs. (news article) |
| DialogSum | En | 13,460 | multiple | daily conversations | ✗[b] | ✗ | abs. |
| AMI | En | 137 | single | meetings | ✓ | ✗ | abs. & ext. |
| ConvoSumm | En | 500 | multiple | online comments | ✓ | ✓ | abs. |
| QMSum | En | 232 | multiple | meetings | ✓ | ✗ | abs. (query-paired) |
| ELITR | 2[c] | 179 | single | meetings | ✓ | ✓ | abs. (minute) |
| VCSum | Zh | 239 | multiple | meetings | ✓ | ✓ | abs. & segmented |
| **ORCHID** | **Zh** | **1,218** | **multiple** | **competitive debates** | ✓ | ✓ | **abs. & stance-based** |

Table 1: Comparison of some existing dialogue summarization datasets. *Lg.* denotes language: *En* for English and *Zh* for Chinese. *Abs.* and *Ext.* denotes abstractive and extractive summary. *Dialogue Topic* indicates whether each dialogue has headline or title. [a]MSAMSum contains parallel corpora of Chinese, English, French and Russian. [b]Part of DialogSum dialogues are argumentative. [c]ELITR includes 59 Czech and 120 English meetings.

| Dataset | Lg. | # Comment | Target | | | Stance Labels | Content Type | Spoken Language |
|---|---|---|---|---|---|---|---|---|
| | | | # | Type | Dep. | | | |
| SemEval-2016 | En | 4,163 | 5 | Entity | TS | Favor, Against, None | social media posts | ✗ |
| NLPCC-2016 | Zh | 4,000[a] | 5 | Entity | TS | Favor, Against, None | social media posts | ✗ |
| CSD | Zh | 5,876 | 1 | Entity | TS | Favor, Against, Neither | social media posts | ✗ |
| MTSD | En | 4,455 | 4 | Entity | MT | Favor, Against, None | social media posts | ✗ |
| X-stance | 3[b] | 67,271 | 150 | Entity | CT | Favor, Against | website comments | ✗ |
| VAST | En | 23,525 | 5,634 | Entity | CT | Pro, Con, Neutral | news article comments | ✗ |
| Emergent | En | 2,595 | 300 | Claim | TI | Favor, Against, Observe | news articles | ✗ |
| IBM Debater | En | 2,394 | 55 | Claim | TI | Pro, Con | Wikipedia articles | ✗ |
| Perspectrum | En | 11,164 | 907 | Claim | TI | 5 labels[c] | website comments | ✗ |
| IAM | En | 4,890 | 123 | Claim | TI | +1, -1 | Wikipedia articles | ✗ |
| **ORCHID** | **Zh** | **16,529[d]** | **2,436** | **Claim** | **TI** | **Pro, Con, Mixed** | **competitive debates** | ✓ |

Table 2: Comparison of some existing stance detection datasets. *Type* denotes whether the targets are entities or claims. *Dep.* denotes the dependency of target (*TS*: target-specific, *MT*: multi-target, *CT*: cross-target, *TI*: target-independent). [a]NLPCC-2016 consists additional 2,400 unannotated comments. [b]X-stance includes cases in French, German and Italian. [c]Perspectrum has *support*, *mildly-support*, *oppose*, *mildly-oppose* and *not a valid perspective* for labels. [d]14,133 debating utterances plus 2,436 closing remarks.

rarer, datasets of two other dependency types have also been proposed: (3) multi-target (e.g., Sobhani et al. 2017; Barrière et al. 2022) and (4) cross-target (e.g., Vamvas and Sennrich 2020; Allaway and McKeown 2020) as Alturayeif et al.'s (2023) survey curated. We summarised a collection of them in Table 2 for comparison.

Prior studies are primarily on target-specific stance detection, and the sources are centered on social media posts. SemEval-2016 (Mohammad et al., 2016) introduced a stance detection shared task on a dataset of 4,163 tweets. A few more shared task datasets were created (Derczynski et al., 2017; Gorrell et al., 2019). Regarding Chinese datasets, NLPCC-2016 (Xu et al., 2016) presented a shared

task similar to SemEval-2016 and released a dataset 4,000 Chinese microblogs. CSD (Li et al., 2022) is newly released and contains 5,876 labelled website comments in Chinese on COVID-19 vaccination.

**Target-Independent Datasets** Turning now to target-independent stance detection datasets, Ferreira and Vlachos (2016) introduced the 'Emergent' dataset that consists 2,595 comments (news article headlines) on 300 claims. IBM Debater (Bar-Haim et al., 2017) leverages 2,394 Wikipedia articles and labelled their stances on 55 claims. More recently, Chen et al. (2019) collected 11,164 website comments on 907 claims. IAM (Cheng et al., 2022) is also created by sourcing Wikipedia

articles. In addition, Durmus and Cardie (2019) constructed a large English online debating corpus with stance labels. Besides language, our dataset differs from it for having additional 'mixed' stance, stance-specific summaries, and spoken style text (Durmus and Cardie's (2019) online debates are not oral ones but by writing threads).

To the extend of our knowledge, there currently is no Chinese target-independent stance detection dataset available. Also, the source texts of existing stance detection datasets are dominantly of written rather than spoken, so our spoken-style corpus could be a rare supplement to the language resource pool.

## 2.2 Dialogue Summarization Datasets

Dialogues range from daily chitchat to formal debate, and researchers have introduced diverse dialogue summarization datasets. SAMSum dataset (Gliwa et al., 2019) greatly accelerated this field by proposing a large-scale dataset of fictitious chat-dialogues created by linguists. English data resources that leverage other forms of dialogues have also emerged: online forum posts (Khalman et al., 2021), interviews (Zhu et al., 2021), debate evidence documents (Roush and Balaji, 2020), daily conversations (Chen et al., 2021), and screenplay transcripts (Chen et al., 2022).

Although Chinese datasets in dialogue summarization are scarce, some valuable corpus were introduced, including a multilingual fictitious chat dataset derived from SAMSum dataset (Feng et al., 2022); medical conversation (Song et al., 2020); sports live commentaries (Huang et al., 2020); customer service logs (Lin et al., 2021; Zou et al., 2021).

**Argumentative Dialogue Summarization Datasets** To reduce the scope, previous English corpora of argumentative dialogue summarization have focused heavily on meetings. AMI (Janin et al., 2003) and ICSI (Carletta et al., 2005) are two early and widely-used meeting corpora. More recently, ConvoSumm (Fabbri et al., 2021) presents 500 conversations of multiple types: news article comments, discussion forums, community question answering and email threads. QMSum (Zhong et al., 2021) uniquely developed 1,808 query-summary pairs on 232 meetings. Wu et al. (2023) introduced a Chinese dataset on 123 meetings annotated with segment-wise summaries. ELITR (Nedoluzhko et al., 2022) is a minuting

corpus that consists of 120 English and 59 Czech meetings.

Despite the work made so far, there remains great imbalance between English dialogue summarization datasets and non-English ones. We also sense an urgency for expanding the diversity of argumentative dialogue corpora beyond meeting transcripts.

## 3 Creating ORCHID

Having reviewed existing stance detection and dialogue summarization corpora, we determine to construct a new versatile dataset that adapts to both tasks. To this end, we introduce ORCHID (**Or**al **Chi**nese **D**ebate), as the name implies, a corpus that features oral debates in Chinese.

We select oral debates in competition scenario as our source for following reasons: (1) debates are highly argumentative and thematic, and stances of both sides are clearly stated and not subject to change; (2) debate utterances are high-quality in terms of logic and rhetoric (Zhang et al., 2016), and such utterances are much less colloquial than daily conversations while retaining partial oral features; (3) since existing stance detection corpora are predominantly written texts, utterances of spoken styles and expressions, which debates offer, could be a valuable addition.

The construction of our dataset consists four major stages: data collection, ASR-aided transcription, manual annotation, and quality control. We first collect videos from public sources. Next, we employ an ASR system to obtain raw transcripts, followed by automatic labelling stances and manual correction. Finally, we extract and construct stance-specific position statements and conclusive summaries. We also apply several quality control measures during the process.

## 3.1 Data Collection

An pilot screening on publicly accessible data reveals that there is an absence of official transcripts for most debate competitions, so we alternatively utilize original debate videos as our primary sources. From major video sharing platforms where the competition organizers regularly release match videos (see Appendix E), we harvest a total of 1,704 debate videos across 59 competitions that were conducted in Mandarin Chinese. After an elaborate filtering process (see §3.4), we end up retaining videos from 1,218 debates across 30

competitions.

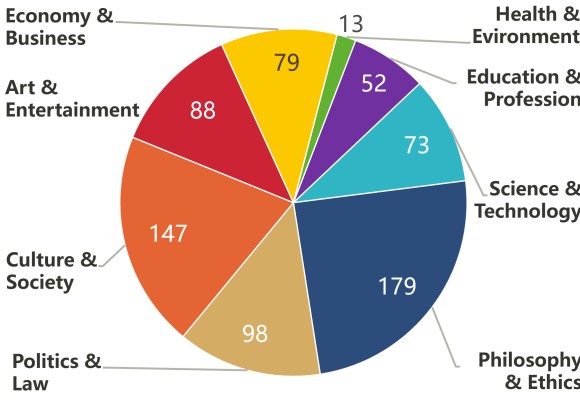

Figure 2: Distribution of topic domains in ORCHID. While there are 476 unique topics, one topic could be classified into multiple domains.

## 3.2 ASR-aided Transcription

We employ a commercially established automatic speech recognition (ASR) system developed by iFLYTEK (see Privacy and Licensing) to obtain raw machine transcription, then our annotators manually post-correct any lexical error or wrongly put punctuation. We generally follow Mirkin et al.'s (2018) pipeline to conduct the ASR-aided transcription from source videos to texts.

To be more specific, firstly, we apply the ASR system on audios to obtain raw machine transcript. Secondly, the contents other than debaters' utterances, such as accidental interruptions and post-debate comments, are discarded. However, we preserve debate adjudicators' announcements, such as pregame introductions and topic statement, for further utilization in the annotation stage. Finally, our annotators manually proofread and correct any lexical and syntactic errors appeared, including wrongly recognized words and misplaced punctuation to create a fully-cleaned transcript.

## 3.3 Automated and Manual Annotation

The annotation consists of four major steps: (1) extracting topics and rephrasing them into position statements, (2) segmenting debates by utterances, (3) labelling utterances with stance and debater, and (4) post-editing transcripts to produce reference summaries.

The annotation is mostly **fact-grounded**, since the truthfulness of topic extraction and utterance labelling can be verified by checking with original videos and hence unanimously agreed upon. We

program scripts to obtain a preliminary segmentation and labelling, a manual correction is followed to correct wrongly split or labelled utterances.

**Topic and Position Statement**  We extract topics from the debate adjudicators' pregame introductions that we retain in the transcription process. In addition, our annotators classify debate topics into eight domains: education & profession, science & technology, philosophy & ethics, politics & law, culture & society, art & entertainment, economy & business and health & environment as shown in Table 2. Since there are topics involving multiple domains, the number of labels is not restricted to one.

Conventionally, debate competitions in English world phrase their topics (or 'motions') into confirmatory assertions (e.g., 'Technology is ethically neutral'), which naturally are position statements. However, most competitions in Chinese world phrase their topics as binary choice questions (e.g., 'Whether technology is ethically neutral?'). Therefore, our annotators manually rephrase question-style topics into position statements for both sides.

**Debate Segmentation**  Debate adjudicators' announcements often contain specific signal phrases (e.g., 'Let us now invite the Third Proposition Speaker to rebuttal') that can serve as round-change indicators. We automated the preliminary segmentation by utilizing the announcements via programmed scripts. Next, the annotators correct missed or wrongly split utterances.

**Stance and Debater**  A typical debate match consists of multiple monologue rounds and discussion (or 'rebuttal') rounds. An utterance in a debate match refers to a piece of time-constrained speech that contains multiple sentences, uttered by either one side (monologue rounds labelled by 'pro' or 'con') or both sides (discussion rounds labelled by 'mixed').

Although we have considered separating the sentences of discussion rounds by pro and con sides, yet we turned down the separation for two reasons: firstly, automatic methods of speaker diarization (Park et al., 2022) yielded unacceptable results (accuracy less than 60%), and a human handpick approach at sentence level would be highly laborintensive; more importantly, we argue that a discussion containing utterances from both sides is a complete linguistic unit. Singling out and concatenating sentences from one side is likely to yield

incoherent and inconsistent contents. Therefore, we keep the integrity of discussion rounds and label them 'mixed'.

Again, the annotators utilize the adjudicators' announcements to label segmented utterances with stance and debater. The announcements are removed after the annotation, retaining only the utterances of debaters.

**Reference Summaries** We construct reference summaries on two levels of granularity: (1) short and concise position summaries; and (2) long comprehensive stance-specific summaries.

The short stance-specific summaries are created by directly adapting position statements. for instance, if the pro side position statement was 'technology is ethically neutral', then the short pro-specific summary would be 'The pro side argues that {pro side position statement}'. By concatenating stance-specific summaries from both side, we get short overall summary: 'The pro side argues that {pro side position statement}, and the con side argues that {con side position statement}.'

The long stance-specific summaries are derived from the closing remarks of debates. At the last round of a formal debate, a debater from each side is expected to provide a summative and comprehensive remark that cover key statements and arguments of their team. we asked our annotators to manually post-edit (e.g., remove greetings and change first-person to third-person) them with minimal changes to the contents (see § 3.4). The long overall summaries are created following the similar pattern as their short counterparts: 'The pro side argues that {pro-specific summary}. The con side argues that {con-specific summary}.'

## 3.4 Quality Control

In order to ensure the quality of the dataset, we filter out videos by following criteria: (1) we drop videos that lack complete debate contents; (2) any video without human-recognizable audio is also discarded because manual post-correction on such video was impracticable; (3) non-standard matches are also neglected (see Appendix A).

During the stance annotation is done independently by two groups of annotators of four (randomly selected from the pool of 12). Members within one group are required to reach unanimous decisions on all instances, so the two groups can be viewed as two collective annotators. Hence, we calculate Cohen's $K$ (Cohen, 1960) to evaluate

inter-annotator agreement. We obtain a Cohen's $K$ close to 1, denoting a very high agreement between the two collective annotators. In fact, there is only 5 differences where the two groups initially disagreed upon, and both groups reached consent after a round of review. The almost perfect agreement is expected since the labelling is fact-grounded.

Regarding the reference summary construction, to avoid personal style preference and bias, we instructed annotators to minimize their editing by limiting changes to removing greetings and shifting addressing to third-person (e.g., from 'we/I/you' to 'the pro/con side'). Additionally, 2 annotators were randomly selected to double-check the correctness and overall consistency of post-editing made by other annotators.

## 3.5 Dataset Overview

The resulting dataset is summarized in Table 3. There are a pair of stance-specific summaries and an overall summary contains both stance-specific summaries for each debate. Hence, 1,218 debates have a total of 2,436 stance-specific summaries and 1,218 overall summaries. It is to be noted that the statistics may be subject to change (see Appendix B).

| Statistics | Value |
|---|---|
| # Total debates | 1,218 |
| # Total unique topics | 476 |
| # Total utterances | 14,133* |
| # Total stance-specific summaries | 2,436 |
| Avg. debate length | 18,349 |
| Avg. # utterances per debate | 11.5* |
| Avg. utterances length | 1,265* |
| Avg. stance-specific summary length | 1,901 |

Table 3: Overview statistics for ORCHID. Debate, utterance and summary average lengths are measured in Chinese characters. *Closing remarks were excluded from calculation.

## 4 Benchmark and Results[2]

Having constructed ORCHID, we conduct an empirical study to benchmark the performances of some existing methods on three challenging tasks against our dataset: (1) stance detection; (2) abstractive summarization; and (3) stance-specific

---

[2]The reported results are based on the 2023-06-15 snapshot of the dataset, hence the statistics in this section does not align with the ones in § 3, for we further expanded and updated the dataset afterwards (see Appendix B).

summarization, a new integrated task that we propose.

To address above tasks, we split our data as summarized in Table 4. For Task 1, a total of 14,091 labelled utterances are split by roughly 78%/11%/11%. To avoid over-fitting, we make sure that utterances from debates with the same topics do not appear in both train and test sets.

|  | Train | Dev. | Test | Total |
|---|---|---|---|---|
| *Stance Detection* | | | | |
| # Pro utterances | 3,321 | 510 | 514 | 4,345 |
| # Con utterances | 3,324 | 511 | 518 | 4,353 |
| # Mixed utterances | 4,360 | 513 | 518 | 5,391 |
| # Total | 11,005 | 1,534 | 1,550 | 14,091 |
| *Abstractive Summarization* | | | | |
| # Overall Summaries | 828 | 104 | 104 | 1,036 |
| *Stance-specific Summarization* | | | | |
| # Pro summaries | 828 | 104 | 104 | 1,036 |
| # Con summaries | 828 | 104 | 104 | 1,036 |
| # Total | 1,656 | 208 | 208 | 2,072 |

Table 4: Data split statistics for benchmark testing.

## 4.1 Task 1: Stance Detection

**Task Definition** Let $D = \{U_i = (c_i, s_i)\}_{i=1}^n$ be a debate of $n$ utterances. Each utterance consists of a piece of text $c_i$ (utterance) and a stance $s_i$. Let $t$ be a designated target claim (a position statement). Given $t$ and $c_i$, the task aims to predict a stance $\hat{s}_i \in \{pro, con, mixed\}$ for each $U$.

**Experiment Setup** As shown in Table 4, stances labels for utterances are imbalanced (31%/31%/38% for 'Pro'/'Con'/'Mixed' of total utterances). Hence, The task is a 3-way classification with imbalanced data, each utterance consisting one single stance label. Following Cheng et al. (2022), we report both overall accuracy and per-class (stance) $F_1$ scores.

We experiment on two well-established pre-trained models: (1) **BERT** (Devlin et al., 2019) and (2) **RoBERTa** (Liu et al., 2019). Specifically, we implement MacBERT-base (Cui et al., 2020), an improved BERT with masked language model (MLM) as correction pre-training task, which mitigates the discrepancy of pre-training and fine-tuning. Regarding RoBERTa, we use RoBERTa-wwm-ext (Cui et al., 2021), a Chinese pre-trained BERT with whole word masking. We fine-tune the models on

the train set, adjusting hyper-parameters using the validation set, run random seeds on the test set 3 times, and report the average results.

Considering that autoregressive LLMs have demonstrated unprecedented performance in many NLP tasks recently, we also devise two direct prompting methods based on **GPT-3.5** (OpenAI, 2022): (3) **Zero-shot Prompting**, a direct prompting method with minimal instructions and (4) **Few-shot Prompting** (Brown et al., 2020) that adds a few utterances (three in our case) with correctly classified stances as examples in prompting (see full prompts can be found in Appendix F). Only the test set is used, and we run 3 times and report the average results.

**Results** As summarized in Table 5, direct prompting methods on autoregressive LLMs (GPT-3.5) outperform fine-tuned bidirectional models, BERT and RoBERTa, by a large margin. This may partially due to the complexity of the task and the limited training data available. In addition, we observe that few-shot prompting boosts GPT-3.5 further on the task. Interestingly, the $F_1$ scores on 'Mixed' label are better than other labels, which may suggest that it is easier to detect conflicting features in an utterance than identifying a particular stance.

| Method | Acc. | $F_1$-Pro | $F_1$-Con | $F_1$-Mixed |
|---|---|---|---|---|
| *Fine-tuning Methods* | | | | |
| MacBERT-base | 56.35 | 50.37 | 49.21 | 50.47 |
| RoBERTa-wwm-ext | 72.13 | 68.36 | 67.18 | 70.84 |
| *Direct Prompting (GPT-3.5) Methods* | | | | |
| Zero-shot | 82.21 | 78.50 | 78.86 | 79.68 |
| Few-shot | **88.71** | **85.77** | **84.72** | **88.13** |

Table 5: Results of target-independent stance detection on ORCHID dataset. *Acc.* denotes overall accuracy.

## 4.2 Task 2: Abstractive Summarization

**Task Definition** Let $D = \{U_1, U_2, \cdots, U_n\}$ be a debate of $n$ utterances (excluding final remarks). The goal of abstractive summarization is to produce a summary $Y_{overall}$ for a given debate $D$.

**Experiment Setup** We follow previous works (See et al., 2017; Gliwa et al., 2019; Roush and Balaji, 2020) to carry out our experiments. We choose the well-established **ROUGE** (Lin, 2004) scores as automatic evaluation metrics and report standard

$F_1$ scores of ROUGE-1, ROUGE-2 and ROUGE-L. We take overall summaries as gold references $Y_{gold\_overall}$. Besides automatic evaluation, we also conduct a human evaluation on generated summaries.

We benchmark two classic extractive methods, three fine-tuning abstractive methods and three direct prompting (specifically `gpt-3.5-turbo-16k-0613`) methods on our dataset. However, our single debate length (18,107 Chinese characters per debate in average) excesses the input size limits of most pre-trained models. We heuristically propose several approaches to address this issue (See full prompts in Appendix F):

- Lead-K: a widely-used simple baseline taking leading $k$ sentences (See et al., 2017). We set $k$ to 3 for benchmarking.

- TextRank-K (Mihalcea and Tarau, 2004): a graph-based ranking model selecting $k$ key sentences based on keyword extraction. We set $k$ to 3 for benchmarking.

- HMNet (Zhu et al., 2020): based on transformer architecture (Vaswani et al., 2017), a hierarchical structure that accommodates long meeting transcripts.

- SUMM$^N$ (Zhang et al., 2022): a multi-stage framework that adapts a coarse-to-fine approach and specialized in handling long input.

- DIALOGLM (Zhong et al., 2022): a pre-trained model that features a window-based de-noising approach. The model also leverage combining sparse attention and conventional attention to process long input.

- Divide-and-Summarize: we prompt GPT-3.5 to obtain individual summary of each utterance, and then integrate summaries to form a complete summary (Koay et al., 2021).

- Accumulative Context Enhanced Divide-and-Summarize: similar to the former method, yet we provide accumulative summary of previous part of the debate as additional context.

- Iterative Revision: first, we prompt the model to obtain the summary of the first utterance, then we provide both previously generated summary and a new utterance and ask the LLM to revise the summary based on the new

utterance. Repeat the process until all utterances are viewed by the model (Zhang et al., 2023).

**Results** We report both automatic metrics and overall human evaluation scores in Table 6. The evaluators were requested to rate the generated summaries by four aspects: conciseness, fluency, faithfulness and informativeness. An overall average is calculated.

We observe that the abstractive methods (HMNet, Summ$^N$ and DIALOGLM) fine-tuned on our dataset generally perform better than the extractive models Lead-3 and TextRank-3. Undoubtedly, the GPT-3.5 direct prompting approach exhibits the most satisfactory performances. Although direct LLM prompting methods yielded better results than fine-tuning methods, there is still large room for improvement. Among three prompting methods, *Iterative Revision* achieve the best overall results.

| Method | R-1 | R-2 | R-L | HE |
|---|---|---|---|---|
| *Extractive Methods* | | | | |
| **Lead-K** | 10.2 | 8.7 | 8.8 | 2.6 |
| **TextRank** | 13.4 | 10.4 | 8.7 | 3.0 |
| *Fine-tuning Abstractive Methods* | | | | |
| **HMNet** | 17.6 | 13.4 | 15.5 | 3.3 |
| **Summ**$^N$ | 20.4 | 15.4 | 18.5 | 3.6 |
| **DIALOGLM** | 19.5 | 10.5 | 14.4 | 3.5 |
| *Direct Prompting (GPT-3.5) Methods* | | | | |
| **Divide-and-Summarize** | 40.6 | 8.3 | 19.2 | 3.8 |
| **A.C.E. Divide-and-Summarize** | 41.1 | 8.0 | 16.9 | 3.9 |
| **Iterative Revision** | 40.4 | 8.6 | 17.8 | 4.3 |

Table 6: Abstractive summarization task results reported in ROUGE scores. **A.C.E.** denotes *Accumulative Context Enhanced*. **HE** denotes overall human evaluation score (1 to 5 rating scale).

### 4.3 Task 3: Stance-specific Summarization

An overall summary of an argumentative dialogue, such as debate or meeting, is time-consuming to comprehend for readers who wish to directly capture stance-specific information. Motivated by this intuition, we propose an integrated task that combines Task 1 and 2.

**Task Definition** Given a debate $D = \{U_i = (c_i, s_i)\}_{i=1}^n$ and a designated stance $s_0 \in \{pro, con\}$. Each utterance consists of a piece of text $c_i$ (utterance) and a stance $s_i$. The task is to (1) produce a stance-specific subset $D_{s_0} = \{U_i =$

$(c_i, s_i = s_0)\}_{i=1}^{k}$; and (2) generate a stance-specific summary $Y_{s_0}$ based on $D_{s_0}$.

**Experiment Setup** For the abstractive methods, we apply a pipeline strategy by first fine-tuning the models (HMNet, Summ$^N$ and DIALOGLM) on stance-specific utterances and their corresponding stance-specific summaries. Next, we ask GPT-3.5 few-shot prompting (for its best performance in Task 1) to distinguish and create a stance-specific subset out of the test set, and then request the models to summarize the utterances. With respect to direct prompting methods, we add simply an instruction asking the model to distinguish stances of given utterances and utilize the ones whose stance match the designated stance (see Table 12).

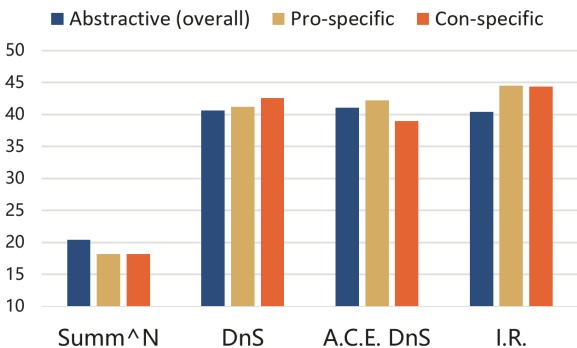

Figure 3: Comparison of R-1 scores of four methods on both overall summarization and stance-specific summarization tasks. DnS: *Divide-and-Summarize*; A.C.E.: *Accumulative Context Enhanced*; I.R.: *Iterative Revision*.

| Method | Stance | R-1 | R-2 | R-L | HE |
|---|---|---|---|---|---|
| *Pipeline Abstractive Methods* | | | | | |
| **GPT-3.5 + HMNet** | Pro | 16.2 | 10.2 | 15.7 | 3.6 |
| | Con | 15.7 | 10.5 | 14.2 | 3.6 |
| **GPT-3.5 + Summ$^N$** | Pro | 18.2 | 14.3 | 16.5 | 3.8 |
| | Con | 18.2 | 13.4 | 17.2 | 3.9 |
| **GPT-3.5 + DialogLM** | Pro | 18.2 | 11.2 | 15.2 | 3.7 |
| | Con | 19.2 | 10.7 | 16.3 | 3.8 |
| *Direct Prompting (GPT-3.5) Methods* | | | | | |
| **Divide-and-Summarize** | Pro | 41.2 | 10.1 | 17.8 | 4.0 |
| | Con | 42.6 | 11.0 | 19.8 | 4.2 |
| **A.C.E.** **Divide-and-Summarize** | Pro | 39.0 | 8.2 | 18.8 | 4.4 |
| | Con | 42.2 | 10.6 | 18.8 | 4.3 |
| **Iterative Revision** | Pro | 44.5 | 11.2 | 19.2 | 4.6 |
| | Con | 44.4 | 11.0 | 17.5 | 4.6 |

Table 7: Stance-specific summarization task results reported in ROUGE scores. **A.C.E.** denotes *Accumulative Context Enhanced*. **HE** denotes overall human evaluation score (1 to 5 rating scale).

**Results** The weak evaluation results, as summarised in Table 7, establish that the dataset is very challenging and the proposed stance-specific summarization task is worthy of future exploration on argumentative texts. End-to-end direct prompting methods are better than the pipeline abstractive methods.

While the comparability between the results from the two summarization tasks should be further examined, we singled out the R-1 scores of Summ$^N$ (for its higher results than the other two abstractive methods) and direct prompting methods in both summarization tasks for a closer comparison as shown in Figure 3. We observe that, for the three abstractive fine-tuning methods (HMNet, Summ$^N$ and DIALOGLM), the stance-specific summarization metrics are lower compared with the ones in overall summarization task. This may due to a non-perfect accuracy in the previous stance detection step. On the other hand, the performances of direct prompting methods are improved (especially *Iterative Revision*). This shows that a preceding highly accurate stance-detection on source text before summarizing could likely to benefit summarization tasks on argumentative dialogues, and an inaccurate one may do the opposite.

## 5 Conclusions

We have presented ORCHID, a novel dataset for target-independent stance detection and argumentative dialogue summarization in Chinese. Our dataset features in rhetorical real-world debates, multi-domain topics, stance-annotated utterances and stance-specific summaries, which invite future utilization to progress various NLP tasks. We benchmark several baseline stance detection and summarization methods on our dataset. Furthermore, we propose a new integrated task that shows potential in improving summarization on argumentative dialogues. Future work can include, for example, devising new models to obtain higher stance-detection accuracy, designing better metrics to better evaluate quality of summarization, and perhaps developing methods to scale up the size and further augment this dataset.

## Limitations

We are aware of some limitations of ORCHID.

**Data Bias**   Admittedly, despite our efforts, potential bias may have been introduced by the demographics of annotators (see Table 8), and we acknowledge it as a limitation of the annotation process. Since the selected debates are drawn from inter-university debate competitions across 2014 to 2023, all utterances were made by students in higher education institutions, and most of the debaters are of East Asian origin.

| Demographic Characteristics | Value |
|---|---|
| Total Participants | 12 |
| Age | [24, 46] |
| Sex (Female/ Male) | 4 / 8 |
| Mandarin Chinese Proficiency | all native |
| Education | undergraduate[a] |

Table 8: The demographic information of the annotators. [a]All evaluators have received at least undergraduate level education.

**Summary Formation**   While we argue that the final remarks of the last round of a debate match in this dataset can serve as comprehensive summaries of the whole debate. First of all, it is advocated by the feature of debate competitions we collected: a debater from each side is expected to provide a summative and comprehensive remark that cover key statements and arguments of their team, and the last round of a match is called the 'Concluding Phase' in those competitions. Secondly, compared to the summaries written by annotators, original remarks from professional debaters are more consistent with the debate contents and strongly stance-based, which are in line with the key stance-specific feature that we wish to highlight in this study.

Nonetheless taking final remarks as summaries has its limitations: in some cases, unseen information or improvised inference were added by the debaters who made the closing remarks. Also, concluding statements in competitive debate scenario are relatively long, resulting low compression ratios. In this dataset, the compression ratios (summary length / debate length) are 0.43 and 0.41 for proposition and opposition side (e.g., SAMsum's one is 0.30).

## Acknowledgement

We appreciate the constructive discussions and insightful comments of all reviewers. We thank Boyu Liu for participating the initial data collection. Spacial thanks to Sihan Zhao from Parsons School of Design for her illustrations in Figure 1.

## Ethics Considerations

### Annotator and evaluator Information

We completed the manual annotation and human evaluation by the collaborators within our research group. The demographic information of them is shown in Table 8.

### Privacy and Licensing

All debate videos we collected were released to public by competition organizers on major video sharing platforms (see Appendix E for details). The ASR system we employed in data construction is the iFLYTEK 2022 real-time ASR package[3] In rare cases, real names of debaters were called during a debate. The annotators were requested to anonymize all personal information during the annotation process.

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

## A   Details about the Selected Debate Competitions

All of selected debate competitions adopt a combination of the British Parliamentary and Australia-Asian debate format with minor variations. Such formats consist four debate phases: opening, rebuttal, free discussion and closing. A group of four debaters make up one side of a debate. Conventionally, the first speaker gives the opening statement; the second speaker and third speaker participates most in rebuttal phase; all members join free discussion, and the forth speaker delivers a closing statement in the end. In all selected debate competitions, there are adjudicators who introduce teams, state topics and rules, maintain order, and control the process of debates.

Competition-specific variations may include: (1) Changing the participants of some sessions: for example, inquiries in rebuttal phase are made by the forth speaker, but some competitions may ask the second speaker to make those inquiries. (2) Adding a 'surprise attack' session: for instance, both side may be given a one-time opportunity to launch a surprise attack between two debate phases, by making either an inquiry or statement. In addition, while the main games of most selected debate competitions takes standard formats, they sometimes host exhibition or entertainment matches that follow special rules.

## B   Reproducibility Statement

Regarding summarization methods we benchmarked, HMNet is publicly available at https://github.com/microsoft/HMNet; SUMM$^N$ is publicly available at https://github.com/psunlpgroup/Summ-N; and DIALOGLM is publicly available at https://github.com/microsoft/DialogLM.

For LLM-prompting methods, we specifically employed the fixed gpt-3.5-turbo-16k-0613 snapshot model for reproducibility purpose, which is available at https://platform.openai.com/docs/models/gpt-3-5/. We kept parameters default, setting temperature to 0.

### Dataset Update

Since we further expanded and updated the dataset after the initial benchmark, statistics are subject to change. Find the up-to-date ORCHID and corresponding statistics at https://github.com/xiutian/OrChiD

| Statistics | Value |
|---|---|
| # Total debates | 1,036 |
| # Total unique topics | 405 |
| # Total utterances | 12,019* |
| # Total stance-specific summaries | 2,072 |
| Avg. debate length | 18,107 |
| Avg. # utterances per debate | 11.6* |
| Avg. utterances length | 1,331* |
| Avg. stance-specific summary length | 1,660 |

Table 9: Overview statistics for the 2023-06-15 snapshot of ORCHID that was tested against in §4. Debate, utterance and summary average lengths are measured in Chinese characters. *Closing remarks were excluded from calculation.

## C  Labels Explanation

| Label | | Meaning |
|---|---|---|
| Stance | PRO | Proposition side |
| | CON | Opposition side |
| | MIXED | Mixed with both sides |
| Debater | ST | Position statement |
| | P# | Debater No. # in proposition side |
| | C$ | Debater No. $ in proposition side |
| | P#C$ / C$P# | Discussion between debater No. # in proposition side and debater No. $ in opposition side. |
| | FREE | Free discussion involving all debaters |
| | SUM | Closing statement, summary |
| | SA-ST | Surprise attack by statement |
| | SA-IN | Surprise attack by inquiry |

## D  Human Evaluation

**Metrics**   Following Allen Institute for AI's GE-NIE leaderboard [4], we choose four aspects as human evaluation metrics. The evaluators are requested to rate generated summaries by four aspects: 1) conciseness, 2) fluency, 3) faithfulness (non-hallucination) and 4) informativeness, following a 1 to 5 rating scale. An overall score is calculated by averaging scores across the aspects with the same weights.

**Instructions**   We randomly split the test set into 12 groups of debates, each consisting around 8 debates. Every evaluator is randomly and exclusively assigned one group of debates. For every debate, we ask the evaluators to 1) read the transcript, 2) read the gold reference summaries, 3) read and rate

the generated summaries by aforementioned metrics. The instructions on both summarization tasks are the same.

---

[4]https://leaderboard.allenai.org/genie-xsum/submissions/about

## E   Data Sources

| Competition Name | Competition Year | # Debate Curated | Organizer Official Release Site |
|---|---|---|---|
| 亚太大专华语辩论公开赛
Asia-Pacific Intervarsity Chinese Debate Tournament | 2013-2023 | 176 | https://www.youtube.com/@user-ws3sj4ui3s
https://space.bilibili.com/442165426 |
| 华语辩论世界杯
Chinese Debate World Cup | 2018-2022 | 321 | https://space.bilibili.com/326420434 |
| 国际华语辩论邀请赛
International Chinese Debating Competition | 2014-2023 | 539 | https://space.bilibili.com/257958427 |
| 世界华语辩论锦标赛
The World Mandarin Debating Championship | 2018-2023 | 151 | https://space.bilibili.com/387099986 |
| 「星耀大湾」国际华语辩论邀请赛
Stars of the GBA International Chinese Debate Tournament | 2022 | 31 | https://space.bilibili.com/13829964 |

Table 10: Overview of data sources. All collected debate videos were released by the official accounts of debate competition organizers at public accessible video sharing platforms including YouTube and Bilibili.

## F   Prompts

### Task 1: Stance Detection

| Method | Prompt |
|---|---|
| Zero Shot | Determine the stance of following utterance and output PRO (proposition side), CON (opposition side) or MIXED (mixed with both sides).
Target Claim: {position_statement}
Utterance: {utterance}
Stance: |
| Few Shot | Consider following examples:

Target Claim: 允许公权力制约假新闻是破坏新闻自由(Allowing public power to restrict fake news is undermining press freedom.*)
Utterance: 我方认为公权力制约假新闻在激励和操作层面有普遍的风险性，因此干扰到新闻人及其发布的内容，即会破坏新闻自由。(We believe that there is a general risk in the incentive and operation of public power restricting fake news, so interfering with journalists and their published content will destroy the freedom of the press.*)
Stance: PRO

Target Claim: 单一货币政策对欧元国家弊大于利(Single monetary policy does more harm than good for euro countries *)
Utterance: 只有引入更多的竞争，才能够促进欧洲更多的产业进行合理的调整之后，让优势的产业代替弱势的产业，才能让欧洲的每个国家得到更好的发展。
(Only by introducing more competition can we promote more industries in Europe to make reasonable adjustments, and let advantageous industries replace weak industries, so that every country in Europe can develop better.*)
Stance: CON

Target Claim: 群众应该要求政治领袖成为道德楷模(The masses should demand that political leaders become moral models.*)
Utterance: 我想请问一下您方定义一下有道德的人和道德的楷模有没有区别？(I would like to ask if there is any difference between your definition of a moral person and a moral model.*)
Stance: MIXED

Determine the stance of following utterance and output PRO (proposition side), CON (opposition side) or MIXED (mixed with both sides).
Target Claim: {position_statement}
Utterance: {utterance}
Stance: |

Table 11: Stance detection task prompts for LLM prompting methods. *Actual prompts do not include English translated text.

## Task 2: Abstractive Summarization

| Method | Prompt |
|---|---|
| Divide -and -Summarize | You are tasked to summarize an utterance from a formal debate. All your responses should be in Chinese. Utterance: {utterance} |
| A.C.E. Divide -and -Summarize | You are tasked to summarize an utterance from a formal debate. Here is also a summary of the previous part of the debate for your reference. All your responses should be in Chinese. Utterance: {utterance} |
| Iterative Revision | Here is the summary of the previous part of a formal debate. You are tasked to revise the summary utilizing the following utterances. All your responses should be in Chinese. Utterance: {utterance} |

Table 12: Abstractive summarization task prompts for LLM-based methods.

## Task 3: Stance-specific Summarization

| Method | Prompt |
|---|---|
| Divide -and -Summarize | You are tasked to summarize an utterance from a formal debate. First determine the stance of the following utterance and output PRO (proposition side), CON (opposition side) or MIXED (Mixed with both sides). If the stance matches {stance}, summarize from the perspective of the {stance} side and return the summary. If the stance does not matched {stance}, return "skipping". All your responses should be in Chinese. Utterance: {utterance} |
| A.C.E. Divide -and -Summarize | You are tasked to summarize an utterance from a formal debate. First determine the stance of the following utterance and output PRO (proposition side), CON (opposition side) or MIXED (Mixed with both sides). If the stance matches {stance}, summarize from the perspective of the {stance} side and return the summary. If the stance does not matched {stance}, return "skipping". Here is also a summary of the previous part of the debate for your reference. All your responses should be in Chinese. Utterance: {utterance} |
| Iterative Revision | Here is the summary of the previous part of a formal debate. You are tasked to revise the summary utilizing the following utterances. First determine the stance of the following utterance and output PRO (proposition side), CON (opposition side) or MIXED (Mixed with both sides). If the stance matches {stance}, summarize from the perspective of the {stance} side and return the revised summary. If the stance does not matched {stance}, do not change the summary. All your responses should be in Chinese. Utterance: {utterance} |

Table 13: Stance-specific summarization task prompts for LLM-based methods.

# G  A Detailed Example of the Dataset

| Stance | Debater | Statement |
|---|---|---|
| **Topic** | | 技术是/不是道德中立的
Whether technology is ethically neutral? * |
| **Pro Statement** | | 技术是道德中立的(Technology is ethically neutral. *) |
| **Con Statement** | | 技术是道德中立的(Technology is not ethically neutral. *) |
| Stance | Debater | Statement |
| PRO | P1 | 探讨技术是否中立，就要看技术有无偏好，能否在此偏向性下产生明确的差异性结果。[...]
Exploring whether technology is ethically neutral requires looking at whether the technology has a bias and whether such bias lead to clear differential outcomes. [...] * |
| MIXED | P1C4 | 没有立场和是中间立场有没有区别？
我方觉得没有立场，不偏向于任何一方，就是一种中立的体现。[...]
Is there any difference between having no position and taking a neutral position?
We feel that having no position and not favoring either side is a manifestation of neutrality. [...] * |
| CON | C1 | 相信在场各位都很清楚，技术作为没有自我意识的非生命体，既无法选择中立，也无法选择倾向。[...]
It is clear to everyone here that technology, as an unconscious non-living entity, cannot choose to be neutral or biased. [...] * |
| MIXED | P4C1 | 换言之好人用就是好技术，坏人用就是坏技术。[...]
In other words, good technology is used by good people and bad technology is used by bad people. [...] * |
| PRO | P2 | 我问你今天我手上如果有一把刀，刀是善的还是恶的？刀既可以用来杀人，医生也可以用来用它来救人，您方的定性要怎么完成？[...]
(Eng. translation) If I had a knife in my hand right now, would it be a good or evil one? The knife can be used to kill people, but doctors can also use it to save people, how do you evaluate it? [...] * |
| MIXED | P2C3 | 所以您方说天地不仁，以万物为刍狗，天地根本就不知道这个世界上发生了什么悲喜。[...]
So you say Heaven and Earth are not benevolent, treating all things as cattle and dogs, and Heaven and Earth don't even know what joys and sorrows are happening in this world. [...] * |
| CON | C2 | 因为任何一项技术创造的时候，人类都有附加的价值和偏好放在上面，这也是它最终向我们传递和引导的结果。[...]
Whenever a technology is created, human values and preferences are placed above it, which are ultimately informative and suggestive to us. [...] * |
| MIXED | P3C2 | 可是锋利的刀杀人会更快，所以还是好的性质吗？在您方体系下杀人是不好的，锋利对刀是好的。[...]
But wouldn't a sharp knife kill faster, so is it still a good quality? Killing is not good under your evaluation, but sharpness of the knife is good. [...] * |
| CON | C3 | [...] |
| PRO | P3 | [...] |
| MIXED | FREE | [...] |
| PRO | **SUM** | 技术本来就是一体两面的，我们在谈论一个技术的时候，是人的善恶让它有了价值的区分，而不是技术本身。所以我们尊重技术，我们不应该以自己的善恶喜好去评价技术。[...]
Technology is a two-sided coin. When we talk discuss a technology, it is people's good and evil that give the technology ethical values, not technology itself. Therefore, we should respect technology and should not judge it by our personal likes and dislikes. [...] * |
| CON | **SUM** | 今天对方辩友问我们，是不是万事万物我们都要给他一个立场，都要给他一个定义？今天我们在这里勇敢地认下来，是这样的。[...]
Today, our opponents' debater asked us if we had to give everything a stance and definition. Here today, we bravely affirm that it is indeed so. [...] * |

Table 14: An example of ORCHID dataset. One entry consists of 1) debate topic, 2) position statements from both sides, 3) utterances labelled with stance and debater, 4) reference specific-summaries of both side. The labels *PRO* and *CON* indicate proposition and opposition stance respectively, while *SUM* denotes summaries. 1) Statements were marked with *PRO-P#* or *CON-C#* denoting debater No.# in proposition or opposition respectively. 2) Discussion rounds involving both side were labelled *MIXED-P#C$* indicating the debaters who participated the discussion (see Appendix C for more details on labels). *Actual data dose not include English translated text.