# OpenReview forum: "ORCHID: A Chinese Debate Corpus for Target-Independent Stance Detection and Argumentative Dialogue Summarization"
_EMNLP/2023/Conference — EMNLP 2023 Main_

### Official Review · Reviewer_rcmE · 2023-08-01

**Soundness:** 4

**Excitement:**

4: Strong: This paper deepens the understanding of some phenomenon or lowers the barriers to an existing research direction.

**Paper Topic And Main Contributions:**

This paper provides a new Chinese debate dialogue dataset with stance and summary annotation. Apart from the overall summary for the whole debate, each dialogue in this dataset also has a stance-specific summary. The authors then apply some baseline methods (pretrained model based methods and large language model based methods) to the new dataset, report their results and give suggestions for future work.

**Questions For The Authors:**

1.	What is the difference between target-specific and target-independent stance detection datasets? Authors should explain it more clearly in Section 2.1.
2.	How is the stance-specific summary be obtained in detail? Could the last round of the debate fully represent the summary of the whole debate?
3.	Why do you need to annotate the stance of each utterance since each utterance has related debater annotation and the stance could be simply known from the debater’s team.
4.	In Line 417-422, why do you say that all methods yield substantial improvements compared to the baseline method, what does the baseline method represent here?
5.	Why does one statement has two related debater? (such as the second statement in Table 9, the debater is P1C4).

**Reasons To Accept:**

1.	The dataset could greatly push the research on Chinese dialogue summarization and stance detection, and the authors promise to make the dataset available to public.
2.	The experiment is elaborate, including traditional summarization methods, recent long document summarization methods and unsupervised LLM methods.
3.	Summary of existing work is detailed and informative. It could let readers quickly understand the existing research progress of stance detection and dialogue summarization.

**Reasons To Reject:**

1.	Some of the information is not presented clearly, such as the criteria of modifying the last round debate to obtain the summary. Besides the quality control of summary is also not introduced in the paper.
2.	The analysis of experiment results is insufficient, as most of the analysis are focused on comparing ROUGE scores. it would be better to make deeper analysis on the challenges of the provided dataset (e.g., error analysis on existing methods).
3.	The authors only introduce a pipeline system to generate stance-specific summary, an end-to-end supervised method may perform better on this specific task.

**Reproducibility:**

4: Could mostly reproduce the results, but there may be some variation because of sample variance or minor variations in their interpretation of the protocol or method.

**Reviewer Confidence:**

4: Quite sure. I tried to check the important points carefully. It's unlikely, though conceivable, that I missed something that should affect my ratings.

---

> ### Author Rebuttal · Authors · 2023-08-28
>
> Thank you for your positive and constructive comments! We are glad that you found ORCHID promising and our experiments elaborate. Below are our responses to specific comments.
>
> ***
>
> >What is the difference between target-specific and target-independent stance detection datasets?
>
> Stance detection aims to classify the stance (‘favor’, ‘against’ and ‘none’) of a piece of comment with respect to a target entity or claim. As we discussed in [Lines 66-72], conventionally the targets of *target-specific* stance detection are **explicit entities** (e.g., ‘Metaverse’), while the targets of *target-independent* stance detection are **claims in the form of complete sentences** (e.g., ‘The commercial value of metaverse is overestimated’) [1, 2, 3]. In our case, three elements in ‘target – comment – stance’ triplet in stance detection task are ‘debate topic – utterance – stance’ correspondingly. The debate topics, or proposition statements are complete sentences, making the dataset to be a *target-independent* one.
>
> [1] Ferreira and Vlachos. 2016. Emergent: A Novel Data-set for Stance Classification. ACL 2016.
> [2] Küçük et al. 2020. Stance Detection: A Survey.
> [3] Hardalov et al. 2022. A Survey on Stance Detection for Mis- and Disinformation Identification. NAACL 2022.
> ***
>
> >How is the stance-specific summary be obtained in detail?
> >Some of the information is not presented clearly, such as the criteria of modifying the last round debate to obtain the summary. Besides the quality control of summary is also not introduced in the paper.
>
> As briefed in Section 3.3, we constructed reference summaries on two levels of granularity, and three summaries (pro-specific, con-specific, overall) were created on each level.
>
> The short stance-specific summaries were created by directly adapting *proposition statements*. Taking the excerpt demonstrated in Table 9 for instance, the pro side proposition statement is ‘technology is ethically neutral’, so the short pro-specific summary is ‘The speaker argues that [pro side proposition statement]’. The short overall summaries are created by concatenating proposition statements from both side: ‘The pro side argues that [pro side proposition statement], and the con side argues that [con side proposition statement].’
>
> The long summaries were derived from the *final remarks* of debates. At the last round of a formal debate, a debater from each side is expected to provide a thematic and comprehensive remark that cover key statements and arguments of their team. As we stated in Section 3.3, we asked our annotators to manually post-edit them with minimal changes to the contents. To be more specific, in order to control quality and avoid any personal style preference and bias, two out of twelve annotators were *randomly selected* to double-check the correctness and overall consistency of post-editing, while the remaining ten annotators were provided with debate transcripts and instructed to edit the final remarks of each side into stance-specific summaries (the original instructions have been added in the appendix in the revised version). We instructed annotators to **minimize their editing**, limiting to removing greetings and changing addressing to third-person (e.g., from ‘we/I/you’ to ‘the pro/con side’). The long overall summaries are created by concatenating stance-specific summaries from both sides: ‘The pro side argues that [pro-specific summary]. The con side argues that [con side summary]’.
>
> ***
>
> >Could the last round of the debate fully represent the summary of the whole debate?
>
> We argue that the final remarks of the last round of a debate match in this dataset can serve as comprehensive summaries of the whole debate. First of all, as we stated in [Lines 327-330], it is advocated by the feature of debate competitions we collected: a debater from each side is expected to provide a thematic and comprehensive remark that cover key statements and arguments of their team, and the last round of a match is called the *‘Concluding Phase’* in those competitions. Secondly, compared to the summaries written by annotators, original remarks from professional debaters are more consistent with the debate contents and strongly stance-based, which are in line with the key *stance-specific* feature that we wish to highlight in this study.
>
> However, as we discussed in [Limitations, Line 567-576], we were aware of some limitations of creating summaries by adapting final remarks. First of all, in some cases, debaters may add new information or inference that is not presented in previous debate phases, making the concluding remarks not entirely ‘faithful’ to the previous contents. In addition, final remarks in competitive debate scenario are relatively long, resulting low compression ratios. The average compression ratios (summary length / debate length) of debates curated in ORCHID are 0.43 and 0.41 for proposition and opposition side respectively (e.g., SAMsum's compression ratio is 0.30) [1].
>
> [1] Gliwa et al. 2019. SAMSum Corpus: A Human-annotated Dialogue Dataset for Abstractive Summarization.
> ***
>
>
> >Why do you need to annotate the stance of each utterance since each utterance has related debater annotation and the stance could be simply known from the debater’s team.
>
> We agree with you and also consider the stance and debate annotation of utterances to be **fact-based**, since the truthfulness of labelling can be verified by checking with original videos and hence unanimously agreed upon. However, an **automatic extraction from original videos is not accurate nor complete**. To be more specific, as we described in Appendix C, the structure of each debate competition varies. Although we programmed scripts to obtain a preliminary segmentation and labelling of transcripts, a manual annotation is necessary to correct wrongly split or labelled utterances, ensuring a high quality of ORCHID.
>
> ***
>
> >Why does one statement have two related debaters? (such as the second statement in Table 9, the debater is P1C4).
>
> A typical debate match consists of multiple monologue phases and discussion (or ‘rebuttal’) phases. An utterance (used interchangeably with ‘statement’ in this paper) in a debate match refers to a piece of time-constrained speech that contains multiple sentences, uttered by either one side (**monologue** phases of debates labelled by ‘pro’ or ‘con’) or both sides (**discussion** phases of debates labelled by ‘mixed’). The average utterance length of this dataset is 1331.26 in Chinese characters.
>
> Taking the second utterance in Table 9 for instance, this is a discussion phase in which the forth debater of con side rebutted while the first debater of pro side defended, hence its debater is labelled ‘P1C4’ and stance labelled ‘Mixed’, as we explained in the caption of Table 9.
>
> Although we had considered separating the sentences of discussion phases by pro and con sides, yet we turned down the separation for two reasons. On the one hand, automatic methods of *speaker diarization* [1] yielded unacceptable results (accuracy less than 60%), and a human annotation approach separating speakers at sentence level will be highly labor-intensive. On the other hand, more importantly, we realized that each discussion utterance is a **complete linguistic unit**. Separating and concatenating sentences from one side may yield incoherent and inconsistent contents. Therefore, we believe it is more prudent to keep the integrity of each utterance
>
> We have rephrased and further clarified the terms used to be more in line with your comment in our revised version.
>
> [1] Park et al. 2021. A Review of Speaker Diarization: Recent Advances with Deep Learning.
>
> ***
>
> >In Line 417-422, why do you say that all methods yield substantial improvements compared to the baseline method, what does the baseline method represent here?
>
> We thank the reviewer for pointing out this issue. What we mean ‘baseline method’ here is actually the ‘zero-shot prompting’. The original expression in Lines 417-422 should be corrected as ‘The performances of direct LLM (GPT-3.5 Turbo in our experiment) prompting methods, both zero-shot and few-zero ones, yield substantial improvements compared to transformer-based fine-tuning methods including BERT and RoBERTa’. We have rewritten this part to clarify in our revised manuscript.
>
> ***
>
> > The analysis of experiment results is insufficient, as most of the analysis are focused on comparing ROUGE scores. it would be better to make deeper analysis on the challenges of the provided dataset (e.g., error analysis on existing methods).
>
> We agree with you that ROUGE scores have a few limitations evaluating summarization tasks, and we have incorporated a more detailed human evaluation analysis to supplement our paper. As we stated in [Human Evaluation, Lines 1002-1009], we conducted a human evaluation on summarization tasks and requested the evaluators to rate generated summaries by: (1) conciseness, (2) fluency, (3) faithfulness (non-hallucination) and (4) informativeness, following a 1 to 5 rating scale. An overall score is calculated by averaging scores across aspects with the same weights. We have elaborated the human evaluation results in those dimensions on existing methods to provide an in-depth analysis on the challenges of ORCHID in our revised version.
>
> ***
>
> Thank you for all the great questions and suggestions! Please let us know if you have any remaining concerns, or if you would consider updating your evaluation based on our response.

---

### Official Review · Reviewer_B1gR · 2023-08-05

**Soundness:** 4

**Excitement:**

3: Ambivalent: It has merits (e.g., it reports state-of-the-art results, the idea is nice), but there are key weaknesses (e.g., it describes incremental work), and it can significantly benefit from another round of revision. However, I won't object to accepting it if my co-reviewers champion it.

**Paper Topic And Main Contributions:**

The authors propose OrChiD, a novel dataset on argumentative dialogues in Chinese language. In particular, the dataset is proposed to address three tasks: (i) target-independent stance detection; (ii) debate summarization; and (iii) stance-specific summarization. The latter task is one the main contributions of the paper. Experiments on the three tasks with several models (BERT, RoBERTa and GPT-3.5) show that stance detection is the easier task while there is ample room of improvement for summarization tasks.

**Questions For The Authors:**

[Orchid] Is there any particular reason on why OrChiD is in coloured text?

**Reasons To Accept:**

- A novel dataset for stance detection and dialogue summarization in Chinese.

- A unique dataset for stance-specific dialogue summarization

**Reasons To Reject:**

- Lack of dataset analysis and comparison with existing corpora.

- Unclear data collection process

## Recommendation

The proposed dataset is a sound contribution: the domain of competitive debates is a suitable candidate for argumentative dialogues and the newly proposed task of stance-specific dialogue summarization is a valuable resource for dialogue analysis. These contributions are further strengthen by the lack of such resources in Chinese.
Nonetheless, I think there are some flaws that should be addressed. First, the lack of a in-depth analysis of the proposed dataset along with comparisons with existing datasets weakens the main contribution of the paper. Second, the data collection process lacks some details, leading to poor reproducibility. Third, many claims regarding the observed experimental results are rather unclear. Please, see `Comments` for more details. Overall, it is hard to place OrChiD with existing literature except for the different domain and the added stance-specific summarization task.
For this reason, I consider the paper in its current status not suitable for acceptance.

## Comments

[Lack of dataset analysis] When providing a novel dataset, a in-depth analysis of collected data in comparison with existing similar datasets is recommended. In this work, such analysis is missing, making hard to understand the properties of Orchid.
A non-exhaustive list includes:
	- Average number of utterances per dialogue
	- Definition of argumentative dialogue (is it the same of other datasets? Is it grounded on some argumentation theory like [1, 2]?)
	- Dialogue formulation (is is the same of other datasets?)
	- Frequency of mixed-labelled utterances (is the 'mixed' label a property of this particular competitive debate instance?)
	- Average debate length in utterances
	- Average utterance length

[Introduction, Lines 057-072] It is unclear to what extent the challenges described in this paragraph are reflected in the Orchid dataset. Could the authors elaborate more in detail?

[Unavailable data and code] I personally find the unavailability of data and code a strong negative point when evaluating a paper. However, I understand that it may not represent a reason to reject a paper. Does the code include the scripts to collect/process the dataset? Does the data also include audio recordings to, for instance, work with audio modality?

[Data Collection] Several details are missing from the description. First, from which platforms where the original debate videos taken? Which criteria was used to select videos? Which ASR system was used? How many annotators were used, how were these annotators selected and which instructions did they follow (Section 3.2)? How were the eight categories chosen (Section 3.3)? Was the preliminary segmentation of transcripts done manually (Section 3.3)?

[Summaries, Section 3.3, Lines 322-334] It is unclear how the short and overall summary is created alongside stance-specific ones. Could the authors clarify this aspect?

[Quality Control, Section 3.4, Lines 347-355] It is unclear why two groups of annotators are created and if the two groups annotate different data. Could the authors further clarify this aspect? Is the IAA computed on all dataset samples? I would also recommend reporting other IAA metrics like Krippendorff's alpha.

[Experiments] How were the models trained? A single train/dev/test split is not a robust evaluation method, especially with a small-to-medium sized dataset like OrChiD. Moreover, are the reported results an average over multiple seed runs? I would recommend a cross-validation routine.

[Results, Section 4.1, Lines 411-422] I'm not sure I understood the paragraph. What do the authors mean with 'baseline method'? Besides, at Lines 416-417 BERT and RoBERTa are distinguished from LLM-based models, while this is not the case at Lines 418-420.

[Task Definition, Section 4.3, Lines 504-506] It is unclear whether the whole dialogue is needed to produce a stance-specific summary. Could the authors further clarify how each gold stance-specific summary is defined by debaters?

[Section 4.3, Lines 520-521] Could the authors further clarify on the choice of Summ^N? Its performance is slightly better than HMNet and on par with DialogLM.

[Results, Section 4.3, Lines 517-537] It is unclear whether the results of Section 4.2 and Section 4.3 are comparable (Lines 522-528). To the best of my understanding the summaries in the two sections are quite different. Additionally, it is unclear whether the presence of a stance-specific summary is specific to the selected competitive debate setting or not. The claim at Lines 535-537 is vague and potentially in contrast with the claim at Lines 491-496.

## Typos and Writing

[Abstract, Line 020] 'propose an integrate summarization task' -> 'propose an integrated summarization task'

[Introduction, Lines 086-087] 'most existing stance detection are' -> 'most existing stance detection datasets are'

[Related Work, Lines 130-131] 'we observe a major amount imbalance'

[Section 2.2, Lines 200-201] 'summarizaiton' -> 'summarization'

[Section 3.1, Lines 267-268] 'see 3.5' -> 'see 3.4'

[Section 3.2, Lines 286-289] I suppose that this task is performed by the hired annotators and not the authors.

[Section 3.3, Lines 318-321] I suppose that this task is performed by the hired annotators and not the authors.

[Section 3.3, Line 323] I would remove 'aforementioned' to make the paragraph self-contained.

[Table 3] Is not referenced in any part of the paper.

[Table 3] '# Total Uniuqe Topics' -> '# Total Unique Topics'

[Section 3.4, Line 354] I would replace 'suggesting' with 'denoting' since a Cohen's K close to 1 is almost perfect agreement.

[Section 4.2, Line 459] 'that adpats a corase-to-fine' -> 'that adapts a coarse-to-fine'

[Figure 3] I would remove 'w/, w/o denotes `with` and `without`'

## References

[1] Douglas Walton. 2008. Informal Logic: A Pragmatic Approach, 2 edition. Cambridge University Press.

[2] Douglas Walton and Fabrizio Macagno. 2007. Types of dialogue, dialectical relevance and textual congruity. Anthropology and Philosophy, 8(1-2):101–120.

**Reproducibility:**

3: Could reproduce the results with some difficulty. The settings of parameters are underspecified or subjectively determined; the training/evaluation data are not widely available.

**Reviewer Confidence:**

4: Quite sure. I tried to check the important points carefully. It's unlikely, though conceivable, that I missed something that should affect my ratings.

---

> ### Author Rebuttal · Authors · 2023-08-29
>
> Thank you for your detailed and constructive comments! We appreciate your recognition of our contribution on introducing a novel dataset and proposing stance-specific summarization task. Below are our responses to specific comments. We number comments and group related ones for cross-reference and readability.
>
> ***
> > 1 When providing a novel dataset, an in-depth analysis of collected data in comparison with existing similar datasets is recommended.
>
> In Table 1 and 2, we have provided comparisons of some key properties of ORCHID with existing related datasets. We have also discussed related datasets and major gaps that motivated us to create ORCHID in \S Related Work: Existing Datasets.
>
> >1.1 Average number of utterances per dialogue / Average debate length in utterances / Average utterance length
>
> We have provided an **overview statistics** that include total number of utterances, sentences and debates of ORCHID in **Table 3**. A typical debate match consists of multiple monologue phases and discussion (or ‘rebuttal’) phases. An utterance (used interchangeably with ‘statement’ in this paper) in a debate competition refers to a piece of time-constrained speech that contains multiple sentences, uttered by either one side (monologue phases of a debate labelled by ‘pro’ or ‘con’) or both sides (discussion phases of a debate labelled by ‘mixed’). The time length of an utterance is determined by competition procedure and monitored by debate adjudicators. The **average number of utterans per dialogue (also the **average debate length in utterances** in our case) is **13.60**, while the **average utterance length** is **1331.26** in Chinese characters.
>
> We have augmented the overview statistics part in our revised version to be more in line with your comments.
>
> >1.2 Definition of argumentative dialogue (is it the same of other datasets? Is it grounded on some argumentation theory like [1, 2]?)
>
> Thank you for this informative comment. Our definition of argumentative dialogue is in line with some previous related work and datasets [1,2,3,4,5]. While argumentation theory does not lie in the primary scope of this paper, we will incorporate the references you mentioned and actively consider exploring this with ORCHID in future work.
>
> [1] John Lawrence and Chris Reed. 2019. Argument Mining: A Survey.
> [2] Farag et al. 2022. Opening up Minds with Argumentative Dialogues.
> [3] Cheng et al. 2022. IAM: A Comprehensive and Large-Scale Dataset for Integrated Argument Mining Tasks.
> [4] Zong et al. 2021. ConvoSumm: Conversation Summarization Benchmark and Improved Abstractive Summarization with Argument Mining.
> [5] Durmus et al. 2019. A Corpus for Modeling User and Language Effects in Argumentation on Online Debating.
>
> >1.3 Dialogue formulation (is the same of other datasets?)
>
> The dialogue formulation in our dataset is in accord with formal debates. To the best of our knowledge, there currently is no such corpus existing in Chinese. Please kindly find an example of our debates in Table 9.
>
> >1.4 Frequency of mixed-labelled utterances (is the 'mixed' label a property of this particular competitive debate instance?
>
> As pointed out in [Line 385], the ‘Mixed’ label utterances take **38%** of total utterances across 1036 debates. As compared to other stance detection datasets in Table 2, to the best of our knowledge, ORCHID is the sole dataset with the ‘Mixed’ label.
>
> ***
>
> >2 [Introduction, Lines 057-072] It is unclear to what extent the challenges described in this paragraph are reflected in the ORCHID dataset. Could the authors elaborate more in detail?
>
> The debates in ORCHID are naturally abundant of *contradictory utterances with conflicting stances*. Consequently, we proposed the stance-specific summarization task to reflect this challenge. The next challenge with respect to stance detection is reflected by that the targets of stance detection in ORCHID are complete claims, making the task to be a *target-independent* one. The other challenge, oral features and noises, while not a primary focus of this dataset, can be investigated in future studies since we will provide the audio tracks from collected videos along with raw and cleaned transcripts, as demonstrated in Table 10.
>
>
> ***
>
> >3 Does the code include the scripts to collect/process the dataset? Does the data also include audio recordings to, for instance, work with audio modality?
>
> In fact, we are more than happy to share the dataset with research community! As we stated in the abstract [Line 24] and introduction [Line 125], we will **release all data and code to public after the anonymity period**, in accord with the anonymity policy of EMNLP. The dataset repo links will be added in the revised version.
>
> Regarding details about files and codes to be released, as shown in Appendix B Table 10, the data files we plan to release include **raw audio recording files, ASR-transcribed text, manual-cleaned text and annotated text**. **Data processing scripts** will be provided along with the data.
> ***
>
> >4 From which platforms where the original debate videos taken? Which criteria was used to select videos? Which ASR system was used?
>
> The platforms that we harvested from include three major video sharing platforms in China: Tencent Video, bilibili and Youku; we also taken some videos from YouTube. We employed the 2022 version of the ASR system developed by iFLYTEK under a commercial-use license, as we indicated in [Privacy and Licensing].
>
> ***
>
> >5 How many annotators were used, how were these annotators selected and which instructions did they follow (Section 3.2)? How were the eight categories chosen (Section 3.3)? Was the preliminary segmentation of transcripts done manually (Section 3.3)?
>
> As shown in Table 8, a total of *twelve* annotators were employed in this study. we selected persons with sufficient Chinese language skills and limited the pool of collaborators to persons whose fields do not overlap with this study to *avoid any conflict of interest*. Please see our response to Comment 6 for more annotation details.
>
> The eight categories were chosen by two steps. Firstly, we brainstormed ten common topic categories based on an overview of the dataset. Then, we mapped each debate to one or more categories, and added new ones if a debate did not fall into any initial category. A total of 18 categories were created. We combined related categories and finalized the number to eight for a balance between informativeness and simplicity.
>
> The preliminary segmentation was done automatically and manually checked. As we discussed in Section 3 [Lines 311-321], we leverage specific signal phrases given by debate adjudicators. We programmed a script to detect those phrases, separate and label debate sections accordingly. Our annotators then manually checked and corrected mislabeled or wrongly-segmented text.
>
> ***
>
> >6 [Summaries, Section 3.3, Lines 322-334] It is unclear how the short and overall summary is created alongside stance-specific ones. Could the authors clarify this aspect?
>
> As we stated in Section 3.3, we constructed reference summaries on two levels of granularity, and three summaries (pro-specific, con-specific, overall) were created on each level.
>
> The short stance-specific summaries were created by directly adapting *proposition statements*. Taking the excerpt demonstrated in Table 9 for instance, the pro side proposition statement is ‘technology is ethically neutral’, so the short pro-specific summary is ‘The speaker argues that [pro side proposition statement]’. The short overall summaries are created by concatenating proposition statements from both side: ‘The pro side argues that [pro side proposition statement], and the con side argues that [con side proposition statement].’
>
> The long summaries were derived from the *final remarks* of debates. At the last round of a formal debate, a debater from each side is expected to provide a thematic and comprehensive remark that cover key statements and arguments of their team. As we stated in Section 3.3, we asked our annotators to manually post-edit them with minimal changes to the contents. To be more specific, in order to control quality and avoid any personal style preference and bias, two out of twelve annotators were *randomly selected* to double-check the correctness and overall consistency of post-editing, while the remaining ten annotators were provided with debate transcripts and instructed to edit the final remarks of each side into stance-specific summaries (the original instructions have been added in the appendix in the revised version). We instructed annotators to **minimize their editing**, limiting to removing greetings and changing addressing to third-person (e.g., from ‘we/I/you’ to ‘the pro/con side’). The long overall summaries are created by concatenating stance-specific summaries from both sides: ‘The pro side argues that [pro-specific summary]. The con side argues that [con side summary]’.
>
> ***
>
> >7 [Quality Control, Section 3.4, Lines 347-355] It is unclear why two groups of annotators are created and if the two groups annotate different data. Could the authors further clarify this aspect? Is the IAA computed on all dataset samples? I would also recommend reporting other IAA metrics like Krippendorff's alpha.
>
> We consider the stance and debate annotation to be **fact-based**, since the truthfulness of labelling can be verified by checking with original videos and hence unanimously agreed upon. The purpose of creating two groups of annotators is to **minimize labelling errors**, and two groups annotated the same whole set of data. As described in Section 3.4, we split eight annotators into two groups of four and required a collective label given by each group on each annotation case (labelling stance and debater). In fact, the 5 differences the two groups initially disagreed from, and both groups reached consent after a collaborative review. We chose Cohen's K because the labels were given by two collective annotators. With just 5 differences present across 14,091 annotation instances, the annotation data yields a **Krippendorff's alpha close to 1** as well.
>
>
> ***
>
> >8 [Experiments] How were the models trained? A single train/dev/test split is not a robust evaluation method, especially with a small-to-medium sized dataset like ORCHID. Moreover, are the reported results an average over multiple seed runs? I would recommend a cross-validation routine.
>
> For some of the methods presented in our paper, no training is required, such as the unsupervised methods Lead-K and TextRank, and the methods based on the OpenAI-provided model, such as Zero-shot & Few-shot GPT-3.5-Turbo Prompting Methods, Divide-and-Summarize, Accumulative Context Enhanced Divide-and-Summarize, and Iterative Revision. For other methods, we trained our models on the train set, adjusted hyperparameters using the validation set, and ran multiple seeds on the test set 3 times, reporting the average results.
>
> Our intention is to provide a set of widely adapted benchmark methods on a unified test set for fair comparison in future research. We appreciate your suggestion for cross-validation and will supplement our results with this approach in the appendix.
>
> ***
>
> >9 [Results, Section 4.1, Lines 411-422] What do the authors mean with 'baseline method'? Besides, at Lines 416-417 BERT and RoBERTa are distinguished from LLM-based models, while this is not the case at Lines 418-420.
>
> We thank the reviewer for pointing out this issue. What we mean ‘baseline method’ here is actually the ‘zero-shot prompting’. The original expression in Lines 417-422 should be corrected as ‘The performances of direct LLM (GPT-3.5 Turbo in our experiment) prompting methods, both zero-shot and few-zero ones, yield substantial improvements compared to transformer-based fine-tuning methods including BERT and RoBERTa’. We have rewritten this part to clarify in our revised manuscript.
>
> ***
>
> >10 [Task Definition, Section 4.3, Lines 504-506] It is unclear whether the whole dialogue is needed to produce a stance-specific summary. Could the authors further clarify how each gold stance-specific summary is defined by debaters?
>
> We argue that the whole dialogue is needed to produce a stance-specific summary. As stated in Section 3.3 and further elaborated in our response to Comment 6, since the gold summaries are adapted from the remarks given by the debaters who had received the whole previous debate contents, it is only fair to define the input of this task the whole dialogue (excluding their final remarks). Please find the detailed process creating reference summaries in our response to Comment 6.
>
> ***
>
> >11 [Section 4.3, Lines 520-521] Could the authors further clarify on the choice of Summ^N? Its performance is slightly better than HMNet and on par with DialogLM.
>
> Our experiment aims to test a versatile set of typical methods of dialogue summarization. Summ^N is chosen because it is a newly-introduced (2022) multi-stage framework for input texts that are longer than the maximum context length of typical pretrained LMs. *Specialization in handling long input* makes it suitable for testing on ORCHID (average debate length of 18,107).
>
> ***
>
> >12 [Results, Section 4.3, Lines 517-537] It is unclear whether the results of Section 4.2 and Section 4.3 are comparable (Lines 522-528). To the best of my understanding the summaries in the two sections are quite different. Additionally, it is unclear whether the presence of a stance-specific summary is specific to the selected competitive debate setting or not. The claim at Lines 535-537 is vague and potentially in contrast with the claim at Lines 491-496.
>
> We thank the reviewer for pointing out this issue. Our intuition for comparing the results between the two tasks to suggest a possibility that, combining stance-specific summaries from both sides may form a better abstractive summary than the one generated without stance classification. We realize it is more comparable that we compare the abstractive summaries generated in Section 4.2 and the combined stance-specific summaries in Section 4.3 with the reference overall summaries. Therefore, we have supplemented the experiment accordingly.
>
> We agree with you that it is yet untested that whether stance-specific summarization is specific to the selected competitive debate setting or not. However, we believe that the proposed task could be generalized to any argumentative text settings where contradictory stances are present and intent to explore this in future work. The primary focus of this work is to provide a versatile testbed for stance detection and dialogue summarization.
>
> What we mean at Lines 491-496 is that although direct LLM prompting methods yielded better results than transformer-based training methods, there is still room for improvement, and our dataset could serve as a testbed for future research improving summarization on argumentative dialogue. We advocate the stance-specific summarization task at Line 535-537. We have rephrased our expressions in our revised version.
>
> ***
>
> > 13 Typos and Writing
>
> Thank you for catching those typos and unclear expressions! We have carefully corrected them. Specifically, regarding [Section 3.2, Lines 286-289] and [Section 3.3, Lines 318-321], you are correct; the manual correction after the automatic was done by the annotators, and we will clarify our expressions in the revised version. In addition, we have added a reference to Table 3.
>
> ***
>
> > 14 [ORCHID] Is there any particular reason on why ORCHID is in coloured text?
>
> Yes! We applied a color scheme (red for pro, blue for con, purple for mixed) on dataset name and Figure 1 to highlight the stance-specific feature of ORCHID. However, we would certainly comply with conference policy if there was any restriction on colored text.
>
> ***
>
> Thank you for all the great questions and suggestions! Hopefully this provides sufficient clarification on our original paper! Please let us know if you have any remaining concerns, or if you would consider updating your evaluation based on our response.

---

### Official Review · Reviewer_LX9i · 2023-08-11

**Typos Grammar Style And Presentation Improvements:** 1) Line 293-294
**Soundness:** 4

**Excitement:**

3: Ambivalent: It has merits (e.g., it reports state-of-the-art results, the idea is nice), but there are key weaknesses (e.g., it describes incremental work), and it can significantly benefit from another round of revision. However, I won't object to accepting it if my co-reviewers champion it.

**Paper Topic And Main Contributions:**

This paper proposes the first Chinese dataset for benchmarking both target-independent stance detection and debate summarization, covering 1036 real-world debates over 403 unique topics. This dataset alleviates this language resource shortage in Chinese, and the authors provide the initial benchmark and present a versatile testbed for future research.

This work focuses on a unique type of dialogue, namely argumentative dialogue, which contains contradictory utterances with conflicting stances and is more complex to be summarized. Further, stance detection in argumentative dialogues is target-independent, where the stances are implicit and should be detected from complete sentences.

Besides these challenges in argumentative dialogue, there still is a lack of multi-domain and annotated Chinese datasets for dialogue (debate) summarization as well as target-independent stance detection. The proposed ORCHID dataset covers these two challenging tasks and enriches the language source for Chinese. The authors also propose a new integrated task, i.e., stance-specific summarization.

**Questions For The Authors:**

1) These debate match videos are collected from video-sharing platforms officially released by competition organizers. May list the sources of these videos in the appendix.

2) Can you discuss any potential biases that may have been introduced by having collaborators within your research group complete the manual annotation and how could you avoid these biases?

**Reasons To Accept:**

1) This paper introduces a novel dataset, ORCHID, for debate summarization and target-independent stance detection, which is the first Chinese dataset for these tasks. The paper also proposes a new integrated task, stance-specific summarization, which is suggested by the experiment results to improve the summarization on argumentative dialogues.

2) During the manual annotation step, the authors sufficiently leverage the features of debate competitions, such as (i) using specific signal phrases as round-change indicators and (ii) using the last round concluding remarks from each side as summaries.

3) The paper conducts preliminary experiments, benchmarking classical and newly-suggested tasks on the proposed dataset, covering a wide range of models. The paper provides a versatile testbed for future research.

4) This paper is well-structured and easy to follow.

**Reasons To Reject:**

1) The limitations of the dataset (as stated in the Limitations section) and potential biases in the annotation and evaluation may affect the validity and generalizability of the results. However, the authors acknowledge these limitations and suggest future work to address them, which may mitigate these risks.


**Reproducibility:**

4: Could mostly reproduce the results, but there may be some variation because of sample variance or minor variations in their interpretation of the protocol or method.

**Reviewer Confidence:**

4: Quite sure. I tried to check the important points carefully. It's unlikely, though conceivable, that I missed something that should affect my ratings.

---

> ### Author Rebuttal · Authors · 2023-08-28
>
> Thank you for your positive comments and constructive feedback! We are glad that you found ORCHID covering these two challenging tasks and enriching existing language source. We appreciate your recognition on our techniques of leveraging features of debates. Below are our responses to specific comments.
> ***
> >These debate match videos are collected from video-sharing platforms officially released by competition organizers. May list the sources of these videos in the appendix.
>
> Sure! We do maintain a list of such, including competition information and video platform link for each debate we collected. We have added the list in the appendix in our revised version and schedule to provide related information when releasing the data.
> ***
> >Can you discuss any potential biases that may have been introduced by having collaborators within your research group complete the manual annotation and how could you avoid these biases?
>
> Thank you for pointing out this issue. We were aware of a few possible biases during the annotation process and had taken measures to minimize them.
>
> In brief, the annotation consists of two major tasks: (1) labelling utterances with stance and debater, and (2) post-editing transcripts to produce reference summaries. We selected 12 persons with sufficient Chinese language skills as the annotators of this study and limited the pool of collaborators to persons whose fields do not overlap with this study to **avoid any conflict of interest**.
>
> We consider the first task to be **fact-based**, since the truthfulness of labelling can be verified by checking with original videos and hence unanimously agreed upon, which left little room for bias. As described in Section 3.4, we split the 8 annotators (**randomly selected** from the pool of 12) into two groups of four and required a collective label given by each group on each annotation case (labelling stance and debater). We calculated Cohen's K [1] of the labels to evaluate inter-annotator agreement and obtained a **Cohen's K close to 1**, denoting a very high agreement. In fact, the 5 differences the two groups initially disagreed from were reviewed by both groups and reached consent.
>
> Regarding the second task, to avoid any **personal style preference and bias**, two out of 12 annotators were **randomly selected** to double-check the correctness and overall consistency of post-editing, while the remaining 10 annotators were provided with ASR-transcribed text and instructed to generate summaries accordingly. Specifically, as described in Section 3.3, the annotators were requested to rephrase the proposition statements and final remarks of each side into stance-specific summaries. We instructed annotators to **minimize their editing** by limiting changes to removing greetings and shifting addressing to third-person (e.g., from ‘we/I/you’ to ‘the pro/con side’). We have added original instructions in the appendix for the revised version.
>
> Admittedly, despite our efforts, potential bias may also have been introduced by the **demographics of annotators** (See Table 8), and we acknowledge it as a limitation of the annotation process.
>
> [1] Jacob Cohen. 1960. Coefficient of Agreement for Nominal Scales.
>
> ***
>
> >Line 293-294: 'In additional,' -> 'In addition,' or 'Additionally,'.
>
> Thank you for catching this typo! We have corrected it in our revised version.
>
> ***
> Thank you for all the great questions and suggestions! Please let us know if you have any remaining concerns, or if you would consider updating your evaluation based on our response.

---

### Meta-Review · Area_Chair_ZL8Q · 2023-09-12

**Recommendation:** 4

**Metareview:**

The authors propose a new Chinese dataset for stance detection, debate summarization, and stance-specific debate summarization for argumentative dialogues. The dataset comes with a set of baseline methods to address the task; They find that summarization performance is relatively low, while stance detection is found to be a relatively easy task.
The reviewers agree that the paper is unique in that it proposes the first Chinese dataset for these tasks, and the modelling experiments provide a good set of baselines on the tasks. One reviewer also highlights the quality of the related work discussion, while another misses a more detailed description of the dataset collection process (this aspect can be addressed easily enough in the final version of the paper, judging from the author rebuttal). The evaluation contains both automatic metrics and human judgments.

---

### Decision · Program_Chairs · 2023-10-07

**Decision:**

Accept-Main

**Comment:**

The authors propose a new Chinese dataset for stance detection, debate summarization, and stance-specific debate summarization for argumentative dialogues. The dataset comes with a set of baseline methods to address the task; They find that summarization performance is relatively low, while stance detection is found to be a relatively easy task.
The reviewers agree that the paper is unique in that it proposes the first Chinese dataset for these tasks, and the modelling experiments provide a good set of baselines on the tasks. One reviewer also highlights the quality of the related work discussion, while another misses a more detailed description of the dataset collection process (this aspect can be addressed easily enough in the final version of the paper, judging from the author rebuttal). The evaluation contains both automatic metrics and human judgments.